# AI-augmented differential diagnosis of granulomatous rosacea and lupus miliaris disseminatus faciei: A 23–year retrospective pilot study

**Sang-Hoon Lee**[1,2], **Hyun Kang**[1], **Seung-Phil Hong**[1], **Eung Ho Choi**[1], **Joong Lee**[3]*, **Minseob Eom**[2]*

**1** Department of Dermatology, Yonsei University Wonju College of Medicine, Wonju, Republic of Korea,
**2** Department of Pathology, Yonsei University Wonju College of Medicine, Wonju, Republic of Korea,
**3** Artificial Intelligence BigData Medical Center, Yonsei University Wonju College of Medicine, Wonju, Republic of Korea

* eomm@yonsei.ac.kr (ME); ljfirst@yonsei.ac.kr (JL)

## Abstract

Granulomatous rosacea (GR) and lupus miliaris disseminatus faciei (LMDF) exhibit overlapping clinical features, making their differentiation challenging. While histopathological examination remains the gold standard, it is invasive and time-consuming, highlighting the need for non-invasive diagnostic approaches. This study evaluates artificial intelligence (AI)-based models for differentiating between GR and LMDF and assess their impact on clinician performance. This retrospective pilot study included 96 patients (62 GR, 34 LMDF) with histopathologically confirmed diagnoses. Neural network models, including convolutional neural networks and vision transformers (ViT), were applied to cropped lesion images while a transformer-based multiple instance learning (TransMIL) approach was used for whole-image analysis. Diagnostic accuracy was also compared between clinicians with and without AI assistance. ViT_base_patch16_224 achieved the highest accuracy (93.0%) and reliability (κ = 0.81) on cropped images, while the TransMIL reached 70% accuracy on whole images. AI augmentation significantly improved clinicians' diagnostic accuracy from 64.7% to 70.3% (p = 0.0136), with the greatest improvement observed among general practitioners. Additionally, mean diagnostic time decreased from 10.7 to 6.4 minutes. These findings highlight the potential of AI models, particularly ViT, in facilitating the differential diagnosis of GR and LMDF. AI-augmented diagnosis improved accuracy and efficiency across all clinician expertise levels, supporting its integration as a complementary tool in dermatological practice.

**Data availability statement:** The data underlying this study include anonymized clinical photographs of patients that cannot be shared publicly due to ethical and privacy concerns. This restriction complies with the data sharing limitations approved by our Institutional Review Board (IRB no. CR324042). Anonymized datasets may be made available upon reasonable request to the corresponding author or to the Institutional Review Board of Yonsei University Wonju College of Medicine (Tel: +82-33-741-1715; Email: wsch.hrpc@gmail.com; Website: https://www.ywmc.or.kr/web/ctm/irb), with appropriate institutional approvals for researchers who meet the criteria for access to confidential data. In accordance with our IRB-approved study protocol, all clinical data will be stored securely on password-protected computers with restricted access for a period of three years. Personal identifiers were converted to coded study IDs prior to analysis, and all data were handled in compliance with institutional policies for anonymization, confidentiality, and restricted data access.

**Funding:** The author(s) received no specific funding for this work.

**Competing interests:** The authors have declared that no competing interests exist.

## 1. Introduction

Granulomatous rosacea (GR) and lupus miliaris disseminatus faciei (LMDF) are rare and chronic granulomatous dermatoses affecting the face [1–3]. Clinically, both diseases are characterized by firm, yellow-brown to erythematous papules or nodules, which often makes clinical differentiation challenging [4]. Unlike GR, LMDF is characterized by the absence of a history of flushing, persistent erythema, or telangiectasia, and is more frequently associated with eyelid involvement, which is uncommon in GR [3,5]. However, the presence of caseating granulomas in histopathological examinations through skin biopsy is essential for differentiating the two conditions [2,3].

Although the association between LMDF and tuberculosis remains controversial [6], there have been rare reports of cases showing similar clinical and histopathological findings to LMDF, where positive results on polymerase chain reaction (PCR) test for *Mycobacterium tuberculosis* were obtained from skin biopsy specimens, followed by subsequent improvement with anti-tuberculous therapy [7,8]. Additionally, we have encountered a case of LMDF associated with a nontuberculous mycobacteria (NTM) infection confirmed by PCR testing (S1 Fig). Therefore, in patients with suspected LMDF, differentiation from GR and evaluation for the presence of mycobacterial infection should be considered.

Recently, the rapid advancement of artificial intelligence (AI) has led to its active application in the field of dermatology [9,10]. Studies have reported the use of AI for the diagnosis and classification of rosacea, demonstrating its ability to classify subtypes of rosacea [11]. However, these studies did not include GR as a subtype of rosacea and primarily utilized the ResNet-50 model. Therefore, in this study, we aimed to apply advanced AI models, including ResNet-50 and the vision transformer (ViT) models, for the differential diagnosis of GR and LMDF. We also compared the accuracy of the AI models in distinguishing GR from LMDF with that of clinicians with different levels of expertise.

## 2. Methods

### 2.1. Study design and participants

We included patients aged 18 years or older who underwent skin biopsies at Yonsei University Wonju Severance Christian Hospital between January 1, 2001, and December 31, 2023, with histopathological findings suggestive of or consistent with GR or LMDF. Subsequently, clinical and histopathological findings were reviewed to distinguish between GR and LMDF, with the presence or absence of caseous necrosis in histopathological examinations serving as the key criterion for differentiation. Patients were excluded if clinical photographs were unavailable or of inadequate quality or if the biopsy site was not located on the face. Additionally, using histopathological specimens, the performance and results of *Mycobacterium* PCR tests, as well as the performance and results of interferon-gamma release assay (IGRA) on blood samples, were evaluated for each patient. Data for this study were accessed on July 1, 2024, for research purposes. Although clinical photographs were included, all images were anonymized by obscuring the eyes and mouth to prevent participant

identification. The authors did not have access to any personally identifiable information of the participants during or after data collection. The study protocol was reviewed and approved by the Institutional Review Board of Yonsei University Wonju College of Medicine (approval no. CR324042). The patients in this manuscript have given written informed consent to the publication of their case details.

## 2.2. Data set

The GR-LMDF dataset consists of a total of 257 image samples, including front-view and both side-view photographs of patients with GR and LMDF (S2 Fig). To ensure anonymization, the eyes and mouth were obscured in all clinical photographs, and only areas containing skin lesions were cropped, resulting in 797 patch images.

## 2.3. Neural network model

In this study, five convolutional neural network (CNN) architectures [12–16] and two ViT models [17] were utilized to classify images of inflammatory skin lesions into two categories, GR and LMDF.

S1 Table provides a brief overview of the deep learning network models used for comparison.

All CNN and ViT models used in this study were initialized with pretrained weights from the ImageNet dataset [18] and subsequently fine-tuned on our GR-LMDF dataset using a transfer learning approach. This method leveraged the general feature representations learned from large-scale natural images while adapting the models to classify inflammatory skin lesions.

A weighted cross-entropy loss function was chosen for this study to address class imbalance [19]. All models were trained for 100 epochs. For CNN architectures, the mini-batch size was set to 24 with a learning rate of 0.001, while for ViT architectures, the mini-batch size was set to 8 with a learning rate of 0.00001. An adaptive moment estimation (Adam [20]) optimizer was used for all models. The models were trained with a fixed patience value of 10, utilizing the ReduceLROnPlateau scheduler (Pytorch [21]) to adjust the learning rate dynamically. The data augmentation techniques, including random rotations within a range of −30 to +30 degrees, horizontal and vertical flipping, and random adjustments to brightness and contrast, were applied to all images in the training set to enhance performance.

## 2.4. Multiple instance learning (MIL)

Training on individual patches achieved an average accuracy of 89%. However, the process of cropping and labelling each patch is not only time-consuming and costly but also differs distinctly from how clinicians typically perform diagnoses. To address this, a transformer-based correlated MIL (TransMIL) approach was adopted to classify GR and LMDF using entire images [22]. This method generates patches from the whole image and learns interactions between patches to predict the label of the entire "bag" [23]. TransMIL utilizes the ViT model to divide images into multiple patches and processes each patch as an independent sequence element. Positional encoding is added to each patch to preserve positional information, which is then fed into the Transformer Encoder. The Transformer Encoder, consisting of multi-head attention mechanisms and feedforward networks, learns the interactions between patches and assigns weights to highlight the most relevant ones. The core of TransMIL lies in its attention-based MIL approach [24], where the relationships between patches are learned, and each patch's contribution to the overall "bag" is weighted. This approach allows the model to emphasize critical patches and ultimately predict the label of the entire image. Additionally, it effectively differentiated GR from LMDF by accurately identifying lesion areas, enhancing classification accuracy for the entire image.

## 2.5. Evaluation

All architectures were compared under identical conditions, with no additional preprocessing or postprocessing applied, except for resizing the input images to the dimensions required by each architecture. For example, all images were provided as RGB color images and resized to match the input dimensions required by each architecture: 224 × 224 for

ResNet-50 and 299×299 for Inception-V3. The dataset was split into 80% for training and 10% each for validation and testing. The model that achieved the best performance during the validation phase was used to evaluate the test set.

The performance of each model was evaluated using metrics such as accuracy, precision, sensitivity, the F1 score, and Cohen's kappa. In this analysis, GR was designated as the negative class, and LMDF as the positive class. Accuracy refers to the proportion of correctly classified cases relative to the total number of cases in the dataset. Precision measures the proportion of positive predictions that were true positives, while sensitivity (or recall) represents the proportion of actual positive cases correctly identified. The F1 score, combining precision and sensitivity, is calculated as their harmonic mean. Cohen's kappa score measures the agreement between evaluators while adjusting for chance agreement [25], providing a more reliable assessment than simple percent agreement. It is particularly useful in scenarios with imbalanced data or multiple categories. Additionally, a confusion matrix was constructed from the results of the testing images to evaluate the performance of the MIL.

### 2.6. Clinician-MIL augmented diagnosis

We compared the accuracy of clinicians with different levels of expertise and that of the MIL model in the differential diagnosis of GR and LMDF. Additionally, we assessed whether clinicians' accuracy and diagnostic time improved when they referred to the MIL-predicted diagnoses on the same test set. The evaluation included three board-certified dermatologists, five dermatology residents, and two general practitioners. In the first assessment (clinical diagnosis), participants were presented with a clinical test set consisting of front-view and side-view photographs (a total of three images per patient) from 16 GR patients and 14 LMDF patients presented in random order. The clinicians were asked to select the most likely diagnosis between GR and LMDF based solely on the clinical photographs.

The second assessment (clinician-MIL diagnosis) was conducted 5 months after the first assessment. Clinicians were provided with the TransMIL model's diagnostic predictions, including its accuracy and confidence levels, for the same clinical test set. Based on this information, clinicians were asked to make their diagnoses. Subsequently, the accuracy and diagnostic time of each clinician from the first and second assessments were compared.

### 2.7. Statistical analysis

The baseline characteristics of the study population are presented as means with standard deviations or frequencies with percentages, according to the type of variable. Differences were evaluated using an independent two-tailed $t$-test or a $\chi^2$ test, as appropriate. Fisher's exact test was used in cases with small sample sizes and frequencies. The results of the first and second evaluations were compared using a paired $t$-test. All statistical analyses were performed using R statistical software (version 3.6.3; R Foundation for Statistical Computing, Vienna, Austria) at a significance level of 5%.

## 3. Results

### 3.1. Study population

A total of 107 patients who underwent skin biopsies were diagnosed with GR or LMDF between January 1, 2001, and December 31, 2023. After applying the exclusion criteria, 96 patients were included in the analysis, comprising 62 GR patients and 34 LMDF patients (Fig 1). The clinical, histopathologic, and laboratory characteristics of the patients are summarized in Table 1. Eyelid involvement was significantly more common in patients with LMDF (70.6%) than in those with GR (29.0%) ($p < 0.001$). Likewise, caseating necrosis was observed in 76.5% of LMDF cases and in none of the GR cases ($p < 0.001$). While LMDF patients demonstrated higher positivity rates for *Mycobacterium* PCR and IGRA tests, these differences were not statistically significant. A total of 171 clinical photographs were obtained from GR patients and 34 from LMDF patients, which were cropped into 797 patch images for use in neural network model training.

**Fig 1. Flowchart of the study.** A total of 62 patients with GR and 34 patients with LMDF were included. The 797 patch images and 257 clinical photographs were utilized for the neural network models and TransMIL, respectively. GR, granulomatous rosacea; LMDF, lupus miliaris disseminatus faciei; TransMIL, transformer based correlated multiple instance learning.

**Table 1. Clinical, Histopathologic, and Laboratory Characteristics of Patients with Granulomatous Rosacea or Lupus Miliaris Disseminatus Faciei.**

| Characteristics | Granulomatous rosacea (n = 62) | | Lupus miliaris disseminates faciei (n = 34) | | p-value |
|---|---|---|---|---|---|
| **Clinical** | | | | | |
| Age, mean (SD), y | 50.5 | (15.5) | 46.6 | (15.6) | 0.244 |
| Sex | | | | | |
| Male | 25 | (40.3%) | 16 | (47.1%) | 0.673 |
| Female | 37 | (59.7%) | 18 | (52.9%) | |
| Eyelid involvement | 18 | (29.0%) | 24 | (70.6%) | <0.001 |
| **Histopathologic** | | | | | |
| Caseating necrosis | 0 | (0.0%) | 26 | (76.5%) | <0.001 |
| **Laboratory** | | | | | |
| *Mycobacterium* PCR | | | | | |
| Tests performed | 21 | (33.9%) | 12 | (35.3%) | 1.000 |
| Positivity | 0 | (0.0%) | †1 | (8.3%) | 0.364 |
| IGRA | | | | | |
| Tests performed | 13 | (21.0%) | 11 | (32.4%) | 0.324 |
| Positivity | 3 | (23.1%) | 4 | (36.4%) | 0.525 |

Table 1 presents the clinical, histopathologic, and laboratory characteristics of the patients with granulomatous rosacea and lupus miliaris disseminatus faciei included in the study. Caseating necrosis was confirmed through review of the histopathology slides.

† This patient was reported as positive for nontuberculous mycobacteria (NTM) and showed complete resolution of the lesions following antibiotic treatment for NTM infection.

Note: IGRA, interferon-gamma release assay; No., number; PCR, polymerase chain reaction; SD, standard deviation.

**Table 2. Accuracy, Precision, Recall, F1 Score, and Kappa Score for the AI Models used in this Study.**

| Models | Accuracy | Precision | Recall | F1 Score | Kappa Score |
|---|---|---|---|---|---|
| ResNet50 | 0.89 | 0.50 | 0.86 | 0.63 | 0.57 |
| Inception-V3 | 0.86 | 0.55 | 0.92 | 0.69 | 0.61 |
| Xception | 0.86 | 0.42 | 0.71 | 0.53 | 0.45 |
| DenseNet169 | 0.89 | 0.50 | 1.00 | 0.67 | 0.61 |
| EfficientNet-B0 | 0.91 | 0.81 | 0.91 | 0.86 | 0.79 |
| ViT_base_patch16_224 | 0.93 | 0.95 | 0.78 | 0.86 | 0.81 |
| ViT_base_patch32_224 | 0.89 | 0.82 | 0.78 | 0.80 | 0.72 |
| †TransMIL | 0.70 | 0.83 | 0.59 | 0.69 | 0.42 |

Table 2 presents the evaluation indices for AI models.

† Unlike other models trained on lesion-specific patches, this approach utilized full-face photographs for differential diagnosis.

Note: AI, artificial intelligence; TransMIL, transformer-based correlated multiple instance learning; ViT, vision transformer.

### 3.2. Comparison of the neural network model

First, we evaluated the ability of each neural network model to differentiate between GR and LMDF lesions. The performance metrics of each model, including accuracy, precision, recall, F1 score, and kappa score, are presented in Table 2. When trained on cropped lesion images, the transformer-based model, ViT_base_patch16_224, demonstrated the highest accuracy at 93.0%. Furthermore, ViT_base_patch16_224 showed the highest performance in both the F1 score, which reflects the balance between precision and recall, and the kappa score, which indicates the reliability of agreement. The transformer-based model, ViT_base_patch16_224, highlights its superior ability to balance diagnostic precision and

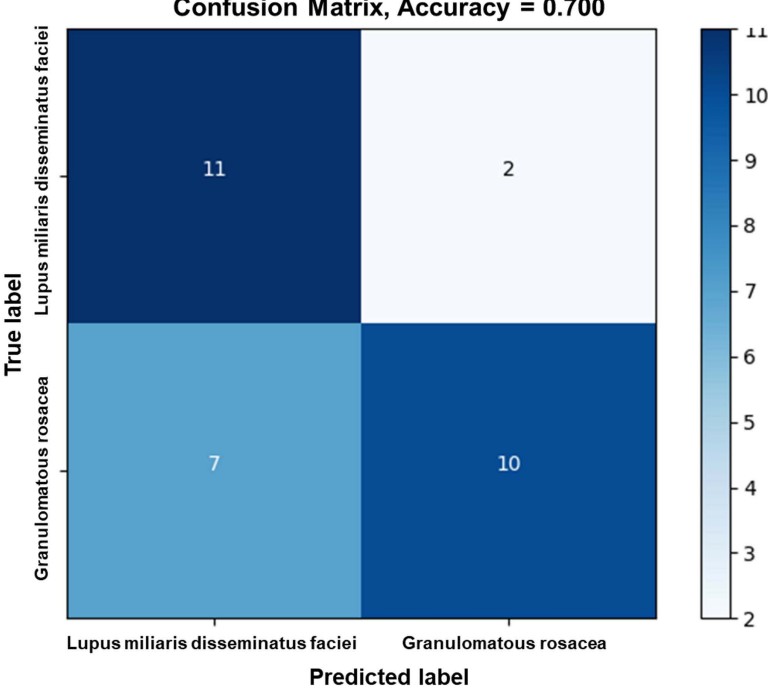

**Fig 2. Performance of multiple instance learning in the differentiation between granulomatous rosacea and lupus miliaris disseminatus faciei.** Confusion matrix shows that the accuracy was 0.700, outperforming the clinician's diagnostic accuracy of 0.647.

consistency compared to other models. Meanwhile, when using the whole image without marking lesion areas and applying a TransMIL model with uniformly sized patches, the accuracy was 70% (Fig 2).

### 3.3. Clinician-AI-augmented diagnosis

The evaluation conducted on a clinical test set comprising 30 patients showed that the mean accuracy for clinical diagnosis was 64.7%±6.3%, whereas clinician-MIL diagnosis achieved a statistically significant improvement to 70.3%±4.7% ($p=0.0136$). When analyzed by expertise level, the mean accuracy for clinical diagnosis was 68.9%±3.8% for board-certified dermatologists, 64.7%±3.8% for dermatology residents, and 58.3%±6.3% for general practitioners. With a clinician-MIL diagnosis, the accuracy improved across all levels of expertise: 72.2%±1.9% for board-certified dermatologists, 69.3%±6.4% for dermatology residents, and 68.3%±4.7% for general practitioners. Notably, the most substantial improvement was observed in general practitioners, who had the lowest level of expertise (Fig 3). The mean diagnostic time decreased from 10.7 minutes in the first assessment to 6.4 minutes in the second assessment, indicating a reduction in diagnostic time with AI assistance ($p=0.1591$).

### 4. Discussion

This retrospective pilot study demonstrated the potential of AI-based approaches, particularly ViT and TransMIL, in differentiating between GR and LMDF. Additionally, we investigated the impact of AI-augmented diagnosis on clinician performance. Our findings revealed several key insights regarding the application of AI in dermatological diagnosis and its potential to augment clinical decision-making.

The superior performance of the ViT_base_patch16_224 model, achieving 93.0% accuracy on cropped lesion images, highlights the effectiveness of transformer-based architectures in dermatological image analysis. This finding aligns with

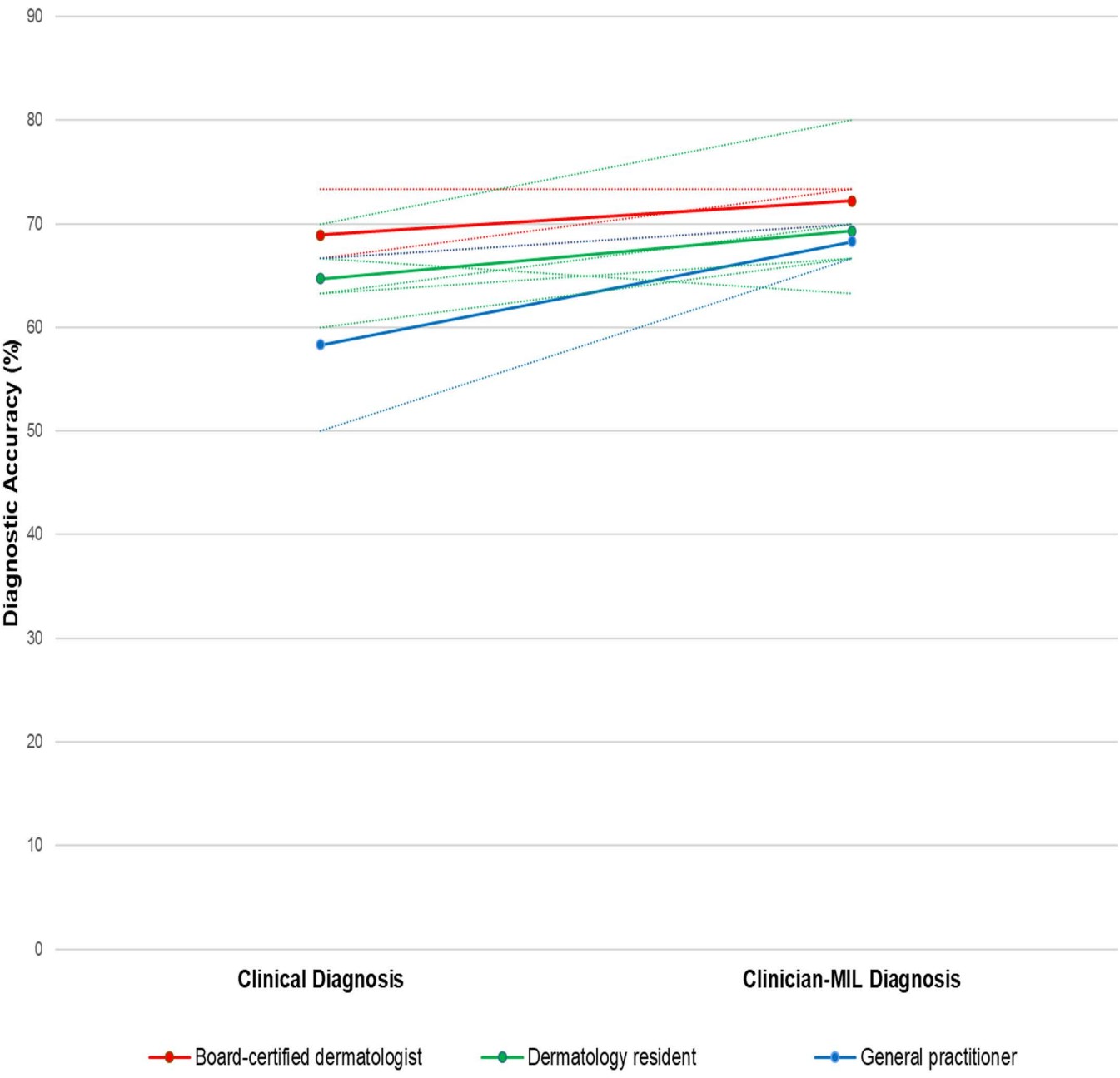

**Fig 3. Clinical diagnosis and clinician-AI augmented diagnosis.** Clinical diagnosis refers to the first assessments made by three board-certified dermatologists (red), five dermatology residents (green), and two general practitioners (blue) on an independent clinical testing dataset. Their second assessment, incorporating assistance from multiple instance learning (MIL), is labelled as the clinician-MIL diagnosis. The dotted lines illustrate the diagnostic accuracy of each clinician during the first and second assessments, while the solid lines represent the average diagnostic accuracy across clinicians with different levels of expertise.

recent literature suggesting that ViTs can capture both local and global features more effectively than traditional CNN architectures [26,27]. The model's ability to maintain high accuracy despite the subtle differences between GR and LMDF lesions demonstrates its potential utility in challenging differential diagnoses. Although caseating necrosis is considered a histopathological hallmark of LMDF, it was observed in 75.6% of LMDF cases in our study. This variability may reflect differences in the stage of the lesion at the time of biopsy, as early or resolving lesions may not display typical caseating necrosis [5]. Such diagnostic uncertainty underscores the clinical value of AI-assisted approaches, which may help overcome the limitations of histopathological interpretation.

However, the decrease in accuracy to 70% when using the TransMIL approach on whole images reflects the challenges of differentiating GR and LMDF, as well as the real-world difficulties of automated dermatological diagnosis. This performance gap between cropped and whole-image analysis suggests that the precise localization of relevant lesions remains a significant challenge in automated dermatological diagnosis. Nevertheless, the TransMIL model's performance matched or exceeded that of clinicians across different expertise levels, indicating its practical utility in clinical settings.

The clinician-MIL augmented approach yielded particularly noteworthy results. The significant improvement in diagnostic accuracy from 64.7% to 70.3% ($p = 0.0136$) when clinicians incorporated AI predictions suggests a synergistic effect between clinician and machine learning capabilities. The most substantial improvement observed among general practitioners (from 58.3% to 68.3%) indicates that AI assistance may be particularly valuable for less experienced clinicians, potentially helping to bridge the expertise gap in dermatological diagnosis.

These findings carry several important clinical implications. First, the ability of AI to improve diagnostic accuracy and efficiency, especially in non-specialist settings, suggests its utility as a triage tool in primary care and teledermatology. Second, improvement in diagnostic accuracy across all expertise levels, combined with the AI's ability to accurately differentiate between GR and LMDF, suggests that AI assistance could enhance the standard of care by supporting timely and appropriate treatment decisions, even in specialist settings. Third, the application of MIL methodologies provides a framework for analysing complex dermatologic images where lesion boundaries are unclear or inconsistently annotated.

In another aspect of this study, we aimed to investigate the association between LMDF and *Mycobacterium* infection, but no statistically significant results were observed. However, in addition to the one case of NTM infection included in the study, there was another case diagnosed as LMDF that was excluded due to poor clinical photograph quality. *Mycobacterium* PCR testing confirmed positivity for *M. tuberculosis* in this case. A previous study on *the Mycobacterium* PCR results for GR and LMDF included only 20 cases of GR and 10 cases of LMDF [3]. In contrast, our study is noteworthy because it included approximately three times as many patients as the previous study. We postulate that a single factor does not cause LMDF but rather represents a stereotypic histological response pattern to a multifactorial aetiology [28–31]. Therefore, in patients suspected of having LMDF, evaluating potential etiologies, including *Mycobacterium* infection, and confirming the findings are crucial for determining the appropriate treatment approach.

This study has several limitations. The single-center retrospective design and the sample size, despite being derived from a 23-year retrospective cohort, remains relatively small, especially for LMDF cases. The class imbalance between GR and LMDF cases, though addressed through weighted loss functions, may have influenced the model's performance. Additionally, the study's focus on Korean patients, all of whom had Fitzpatrick skin phototypes III or IV, may affect the model's applicability to populations with different ethnic backgrounds or skin phototypes. In addition, while AI models performed well in a controlled dataset, real-world variability in image quality and clinical presentation may impact their generalizability. Future studies should aim to validate these findings in larger multicenter cohorts with diverse imaging datasets. Additionally, integrating histopathological features and laboratory findings into AI algorithms may further enhance diagnostic accuracy.

## 5. Conclusion

Our pilot study demonstrates the feasibility and potential utility of AI-based approaches in differentiating between GR and LMDF. The successful implementation of ViT and MIL, combined with the demonstrated benefits of clinician-AI augmentation, suggests a promising path forward in dermatological diagnosis. These findings support the continued development and integration of AI tools in clinical dermatology practice, particularly as aids to clinical decision-making rather than as autonomous diagnostic systems.

## Supporting information

**S1 Table. Details of neural network models used in this study.**
(DOCX)

**S1 Fig. A case of lupus miliaris disseminatus faciei associated with nontuberculous *Mycobacterium* infection.**
(DOCX)

**S2 Fig. Examples of three photos taken from three different angles of a granulomatous rosacea patient.**
(DOCX)

## Author contributions

**Conceptualization:** Sang-Hoon Lee, Eung Ho Choi, Minseob Eom.

**Data curation:** Sang-Hoon Lee, Hyun Kang, Seung-Phil Hong, Eung Ho Choi, Minseob Eom.

**Formal analysis:** Seung-Phil Hong, Joong Lee.

**Investigation:** Sang-Hoon Lee, Hyun Kang, Joong Lee.

**Methodology:** Sang-Hoon Lee, Eung Ho Choi, Joong Lee, Minseob Eom.

**Project administration:** Sang-Hoon Lee, Minseob Eom.

**Resources:** Seung-Phil Hong, Eung Ho Choi, Minseob Eom.

**Software:** Joong Lee, Minseob Eom.

**Supervision:** Seung-Phil Hong, Eung Ho Choi, Joong Lee, Minseob Eom.

**Validation:** Sang-Hoon Lee, Seung-Phil Hong, Joong Lee.

**Visualization:** Sang-Hoon Lee, Joong Lee.

**Writing – original draft:** Sang-Hoon Lee.

**Writing – review & editing:** Sang-Hoon Lee, Hyun Kang, Seung-Phil Hong, Eung Ho Choi, Joong Lee, Minseob Eom.

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
