## [Decision Letter · Decision Letter 0]

Dear Dr. Eom,

Thank you for submitting your manuscript to PLOS ONE. After careful consideration, we feel that it has merit but does not fully meet PLOS ONE’s publication criteria as it currently stands. Therefore, we invite you to submit a revised version of the manuscript that addresses the points raised during the review process.

We look forward to receiving your revised manuscript.

Kind regards,

Aneesh Basheer

Academic Editor

PLOS ONE

Journal Requirements:

3. In the online submission form, you indicated that [The data underlying this study include anonymized clinical photographs of patients, which contain sensitive patient information. Due to ethical and privacy concerns, these data cannot be shared publicly. However, anonymized datasets may be made available upon reasonable request to the corresponding author and with appropriate institutional approvals for researchers who meet the criteria for access to confidential data.]. 

Additional Editor Comments:

Dear authors,

Please revise the paper for minor changes as advised by the reviewers and then your manuscript will be reconsidered.

Reviewers' comments:

Reviewer's Responses to Questions

**Comments to the Author**

1. Is the manuscript technically sound, and do the data support the conclusions?

Reviewer #1: Yes

Reviewer #2: Yes

2. Has the statistical analysis been performed appropriately and rigorously?

Reviewer #1: Yes

Reviewer #2: Yes

3. Have the authors made all data underlying the findings in their manuscript fully available?

Reviewer #1: Yes

Reviewer #2: Yes

4. Is the manuscript presented in an intelligible fashion and written in standard English?

Reviewer #1: Yes

Reviewer #2: Yes

Reviewer #1: This is an interesting and relevant study, especially because of the increasing use of AI in medicine to enhance clinical skills. Dermatology often relies on pattern recognition to make a diagnosis and therefore may benefit significantly by the incorporation of AI. This study effectively demonstrates how AI can augment clinical acumen in differentiating LMDF and granulomatous rosacea which could otherwise pose a diagnostic dilemma. As noted, further research is essential to enhance the generalizability of these findings to various ethnic groups and to refine AI algorithms for different dermatological conditions, thereby improving diagnostic accuracy before integrating it as a part of routine clinical practice.

Both entities can be clinically indistinguishable, and some experts consider LMDF to be a variant of GR. However, eyelid involvement is more commonly seen in LMDF, while it is uncommon in GR - this detail could enhance diagnostic precision and be a valuable addition in machine learning . Furthermore, although caseation necrosis is often considered as a histopathological hallmark of LMDF, it is not always present. The stage of disease at the time of biopsy often influences histological findings - early or resolving lesions may not show classical histopathological findings. This variability often adds to the diagnostic difficulty and strengthens the importance of AI in making an accurate diagnosis. Including these aspects would strengthen the article.

Reviewer #2: 1. How convolutional neural network (CNN) architectures and ViT 87 models were trained before undertaking the study? Kindly include the details

2. Was dermoscopy part of the clinical examination?

3. What was the skin phototype and ethnicity of the patients?

**Do you want your identity to be public for this peer review?** For information about this choice, including consent withdrawal, please see our Privacy Policy

Reviewer #1: No

Reviewer #2: No

---

## [Author Response · Author response to Decision Letter 1]

2 Jun 2025

[30 May 2025]

Aneesh Basheer

Academic Editor

PLOS ONE

Submission of revised manuscript, PONE-D-25-11497_R1: AI-augmented differential diagnosis of granulomatous rosacea and lupus miliaris disseminatus faciei: A 23-year retrospective pilot study

Dear Editor:

We write to acknowledge safe receipt of your email and would like to thank you and the reviewers for the care and effort they have taken in reviewing the manuscript. We have gone over all the points that they raised, and we have uploaded our revised manuscript incorporating responses to the reviewer comments.

Below are the FULL comments we received and the corresponding point-to-point responses.

Review Comments to the Author

Reviewer #1:

This is an interesting and relevant study, especially because of the increasing use of AI in medicine to enhance clinical skills. Dermatology often relies on pattern recognition to make a diagnosis and therefore may benefit significantly by the incorporation of AI. This study effectively demonstrates how AI can augment clinical acumen in differentiating LMDF and granulomatous rosacea which could otherwise pose a diagnostic dilemma. As noted, further research is essential to enhance the generalizability of these findings to various ethnic groups and to refine AI algorithms for different dermatological conditions, thereby improving diagnostic accuracy before integrating it as a part of routine clinical practice.

Both entities can be clinically indistinguishable, and some experts consider LMDF to be a variant of GR. However, eyelid involvement is more commonly seen in LMDF, while it is uncommon in GR - this detail could enhance diagnostic precision and be a valuable addition in machine learning. Furthermore, although caseation necrosis is often considered as a histopathological hallmark of LMDF, it is not always present. The stage of disease at the time of biopsy often influences histological findings - early or resolving lesions may not show classical histopathological findings. This variability often adds to the diagnostic difficulty and strengthens the importance of AI in making an accurate diagnosis. Including these aspects would strengthen the article.

Response: Thank you for this valuable and insightful comment. In response to your suggestion, we conducted a thorough re-analysis of the clinical photographs and histopathology slides of all patients included in the study. Based on this review, we have incorporated the rates of eyelid involvement and caseating necrosis into Table 1 to enhance the diagnostic context. Specifically, eyelid involvement was noted in 24 out of 34 patients with LMDF and in 18 out of 62 patients with GR. Caseating necrosis was identified in 26 out of 34 patients with LMDF. These findings have also been integrated into the main text to underscore their relevance in clinical differentiation, as detailed below:

(Page 3): Clinically, both diseases are characterized by firm, yellow-brown to erythematous papules or nodules, which often makes clinical differentiation challenging [4]. Unlike GR, LMDF is characterized by the absence of a history of flushing, persistent erythema, or telangiectasia, and is more frequently associated with eyelid involvement, which is uncommon in GR [3, 5]. However, the presence of caseating granulomas in histopathological examinations through skin biopsy is essential for differentiating the two conditions [2, 3].

(Page 11): The clinical, histopathologic, and laboratory characteristics of the patients are summarized in Table 1. Eyelid involvement was significantly more common in patients with LMDF (70.6%) than in those with GR (29.0%) (p < 0.001). Likewise, caseating necrosis was observed in 76.5% of LMDF cases and in none of the GR cases (p < 0.001). While LMDF patients demonstrated higher positivity rates for Mycobacterium PCR and IGRA tests, these differences were not statistically significant. A total of 171 clinical photographs were obtained from GR patients and 34 from LMDF patients, which were cropped into 797 patch images for use in neural network model training.

(Page 12): Table 1. Clinical, Histopathologic, and Laboratory Characteristics of Patients with Granulomatous Rosacea or Lupus Miliaris Disseminatus Faciei

Cases, No. (%)

Granulomatous rosacea Lupus miliaris disseminatus faciei p-value

(n = 62) (n = 34)

Clinical characteristics

Age, mean (SD), y 50.5 (15.5) 46.6 (15.6) 0.244

Sex

Male 25 (40.3%) 16 (47.1%) 0.673

Female 37 (59.7%) 18 (52.9%)

Eyelid involvement 18 (29.0%) 24 (70.6%) < 0.001

Histopathologic characteristic

Caseating necrosis 0 (0.0%) 26 (76.5%) < 0.001

Laboratory characteristics

Mycobacterium PCR

No. of tests conducted 21 (33.9%) 12 (35.3%) 1.000

Positivity 0 (0.0%) †1 (8.3%) 0.364

IGRA

No. of tests conducted 13 (21.0%) 11 (32.4%) 0.324

Positivity 3 (23.1%) 4 (36.4%) 0.525

Note: The table shows the clinical, histopathologic, laboratory characteristics of the patients with granulomatous rosacea and lupus miliaris disseminatus faciei included in the study. Caseating necrosis was confirmed through review of the histopathology slides.

(Page 17): The model’s ability to maintain high accuracy despite the subtle differences between GR and LMDF lesions demonstrates its potential utility in challenging differential diagnoses. Although caseating necrosis is considered a histopathological hallmark of LMDF, it was observed in 75.6% of LMDF cases in our study. This variability may reflect differences in the stage of the lesion at the time of biopsy, as early or resolving lesions may not display typical caseating necrosis [5]. Such diagnostic uncertainty underscores the clinical value of AI-assisted approaches, which may help overcome the limitations of histopathological interpretation.

Reviewer #2:

1. How convolutional neural network (CNN) architectures and ViT 87 models were trained before undertaking the study? Kindly include the details

Response: Thank you for pointing this out. All CNN and ViT models employed in this study were initialized with pretrained weights from the ImageNet dataset. We then fine-tuned these models on our GR–LMDF dataset using a transfer learning approach, allowing the networks to adapt their learned visual representations to the specific task of classifying granulomatous rosacea and LMDF lesions. We have now clarified these details in the Methods section to improve transparency and reproducibility.

(Page 6): In this study, five convolutional neural network (CNN) architectures [11-15] and two ViT models [16] were utilized to classify images of inflammatory skin lesions into two categories, GR and LMDF.

S1 Table provides a brief overview of the deep learning network models used for comparison.

All CNN and ViT models used in this study were initialized with pretrained weights from the ImageNet dataset [18] and subsequently fine-tuned on our GR-LMDF dataset using a transfer learning approach. This method leveraged the general feature representations learned from large-scale natural images while adapting the models to classify inflammatory skin lesions.

A weighted cross-entropy loss function was chosen for this study to address class imbalance [17]. All models were trained for 100 epochs. For CNN architectures, the mini-batch size was set to 24 with a learning rate of 0.001, while for ViT architectures, the mini-batch size was set to 8 with a learning rate of 0.00001. An adaptive moment estimation (Adam [18]) optimizer was used for all models. The models were trained with a fixed patience value of 10, utilizing the ReduceLROnPlateau scheduler (Pytorch [19]) to adjust the learning rate dynamically. The data augmentation techniques, including random rotations within a range of -30 to +30 degrees, horizontal and vertical flipping, and random adjustments to brightness and contrast, were applied to all images in the training set to enhance performance.

2. Was dermoscopy part of the clinical examination?

Response: Thank you for raising this question. This study was a retrospective analysis of patients who visited our institution between 2001 and 2023. Clinical data were collected as gross photographs, as shown in Figure S2. Dermoscopic images were not available and were not part of the routine clinical documentation during the study period.

3. What was the skin phototype and ethnicity of the patients?

Response: Thank you for highlighting this. All patients included in this study were ethnically East Asian (Korean), and their Fitzpatrick skin phototypes were either type III or IV based on the review of clinical photographs. This homogeneity of the study population was acknowledged as a limitation in the Discussion, and we have revised it as follows:

(Page 19): “Additionally, the study’s focus on Korean patients, all of whom had Fitzpatrick skin phototypes III or IV, may affect the model’s applicability to populations with different ethnic backgrounds or skin phototypes.”

Sincerely,

Minseob Eom, MD, PhD

Department of Pathology, Yonsei University Wonju College of Medicine

20 Ilsan-ro, Wonju, Gangwon 26426, Republic of Korea

Tel: +82-33-741-1554; Fax: +82-33-741-0423

Email: eomm@yonsei.ac.kr

Joong Lee

Artificial Intelligence BigData Medical Center,

Yonsei University Wonju College of Medicine, Wonju,

2, Galmeori 2-gil, Wonju, Gangwon 26417, Republic of Korea

Tel: +82-33-741-5452; Fax: +82-33-741-5453

Email: ljfirst@yonsei.ac.kr

---

## [Editor Report · Decision Letter 1]

AI-augmented differential diagnosis of granulomatous rosacea and lupus miliaris disseminatus faciei: A 23-year retrospective pilot study

PONE-D-25-11497R1

Dear Dr. Eom,

We’re pleased to inform you that your manuscript has been judged scientifically suitable for publication and will be formally accepted for publication once it meets all outstanding technical requirements.

Kind regards,

Aneesh Basheer

Academic Editor

PLOS ONE

Additional Editor Comments (optional):

Thank you for responding to the reviewer comments.
---

## [Editor Report · Acceptance letter]

PONE-D-25-11497R1

PLOS ONE

Dear Dr. Eom,

I'm pleased to inform you that your manuscript has been deemed suitable for publication in PLOS ONE. Congratulations! Your manuscript is now being handed over to our production team.

Kind regards,

on behalf of

Dr. Aneesh Basheer

Academic Editor

PLOS ONE